# Is Artificial Intelligence an Elixir to the Software Engineering Community? An Empirical Study Among Managers

Xin Zhao
xzhao1@seattleu.edu
Seattle University
Seattle, WA, USA

Brian Vu
bvu1@seattleu.edu
Seattle University
Seattle, WA, USA

Sitesh Pattanaik
siteshpattanaik@gmail.com
Amazon
Seattle, WA, USA

## Abstract

**Background:** Artificial intelligence is rapidly changing the landscape of software development. With the unique ability to quickly generate code and the potential to disrupt traditional workflows, AI tools have found growing adoption within the software development process. Subsequently, this topic has been the focus of academic work, including research examining qualitative impacts to productivity and the analysis of sentiments from the developers who utilize AI tools. While this material is extensive, our research team identified a gap within existing literature: what do software managers have to say? **Goals:** The overarching goal of this study is to examine the views of software managers on how AI tools have affected software development. We seek to understand how managers, who leverage a top-down view of the development process, perceive the influence of AI on developers, their own roles, and the broader labor market. **Methodology:** To answer these questions, we conducted an empirical study by releasing an online questionnaire containing both qualitative and quantitative questions, sampling software managers employed across both tech-focused and non-tech-focused companies. **Results:** Through a survey of 42 managers, we found that managers hold nuanced views on the introduction of AI into software development. They encourage developers to use AI, perceive it as valuable for testing, and apply it themselves for knowledge work. At the same time, they raise concerns about privacy, responsibility, transparency, and over-reliance. Many also predict a loss of jobs within the software development market due to consolidation driven by AI. **Conclusion:** AI is seen by managers as both a powerful productivity tool and a source of new ethical challenges. Our investigation paves the way for a comprehensive understanding of how AI is perceived by those who directly manage the introduction of these tools into traditional software development workflows, revealing a road map for future endeavors for the software development community.

## CCS Concepts

• **General and reference** → **Empirical studies**; • **Computing methodologies** → **Artificial intelligence**; • **Social and professional topics** → **Computing organizations**; *Employment issues.*

## Keywords

Empirical Software Engineering, Artificial Intelligence, Software Development Managers

**ACM Reference Format:**

Xin Zhao, Brian Vu, and Sitesh Pattanaik. 2026. Is Artificial Intelligence an Elixir to the Software Engineering Community? An Empirical Study Among Managers. In *Proceedings of the 3rd ACM International Conference on AI-Powered Software (AIware '26), July 06–07, 2026, Montreal, QC, Canada.* ACM, New York, NY, USA, 10 pages. https://doi.org/10.1145/3805760.3814907

## 1 Introduction

Recently, Artificial Intelligence (AI) has gained prominence in both professional and public discourse. There is a broad trend of AI integration in domains including healthcare [48], education [55], finance [18], agriculture [8], and the creative industries [4]. Notably, this trend has become increasingly prominent within the discipline primarily responsible for its inception: software development [9]. AI tools are currently transforming how software is designed, executed, and maintained. From publicly available AI chat bot services such as ChatGPT [1] to proprietary internal solutions, they enable developers to generate and refactor code rapidly.

As strong as the benefits that AI provides may be, an important question remains: how will traditional software development workflows change and human responsibilities shift? Additionally, a range of factors and concerns influence whether developers are willing to adopt AI tools in their work [28] [54] [16], possibly serving as a reason for hesitation. Accordingly, much academic work has been completed to analyze the influence of AI on software development [35][24][7], attempting to understand the potential benefits and drawbacks of AI integration. However, current literature lacks concentrated empirical research on this issue from the viewpoint of software managers. This perspective is rich with insight as managers occupy a strategic, top-down vantage over the development process. Because they leverage key metrics on software creation and maintenance and shape how their teams integrate AI tools, their perspective is critical.

Thus, given an academic gap on the software manager's perspective, we completed an empirical study via an online questionnaire that surveys 42 software managers. Our survey contained both closed and open-ended questions, seeking the perspective of managers regarding the three following aspects: the impact of AI on software developers, the impact on software managers themselves, and the impact on the job market.

The contributions of our investigation are as follows:

*AIware '26, Montreal, QC, Canada*

© 2026 Copyright held by the owner/author(s).

ACM ISBN 979-8-4007-2601-9/2026/07

https://doi.org/10.1145/3805760.3814907

[1] https://chatgpt.com/

(1) Examination of empirical perceptions of software managers on AI from different dimensions of the software development process.
(2) Guidance for software professionals to navigate the evolving landscape of AI in software development.
(3) Recommendations for the software community to mitigate the ethical concerns associated with AI integration.

The remainder of this paper is organized as follows. Section 2 surveys related literature. Section 3 establishes the theoretical background and defines key terminology. Our research methodology is described in Section 4, followed by discussion of results in Section 5. We examine threats to validity in Section 6 and discuss the broader implications of our findings in Section 7. Finally, Section 8 offers concluding remarks and directions for future work.

## 2 Related Work

Beyond the managerial viewpoint, other key areas of academic research must be examined to understand how AI affects the software development process. Thus, within this section, we examine three related topics that are complementary to the manager's perspective: AI's role within management practices, measuring AI's coding performance, and hesitancy toward new technologies.

### 2.1 AI's Role Within Management Practices

While AI usage in software development generally brings to mind generative tools used to produce and refactor code quickly, the broad umbrella term of "artificial intelligence" also encompasses systems used to support managerial decision-making [50] [5].

Parikh [39] concludes that AI can significantly improve software product management activities, resource usage, product outcomes, and user experiences. Chaturvedi et al. [14] conducted a literature review of articles that highlight AI's ability to impact managerial decision-making, stressing "human-AI collaboration's importance in reducing cognitive biases like anchoring and overconfidence."

In contrast, Cao et al. [11] noted doubts about utilizing AI to improve organizational decision-making. They recommend implementing a mechanism to alleviate managers' personal concerns about the tool, placing an emphasis on the hidden biases of AI.

The application of AI for management purposes extends beyond the scope of software managers. Abositta et al. [1] highlighted "the potential for technological integration to enhance organizational strategic capabilities" in "manufacturing, construction, and information technology firms".

### 2.2 Measuring AI's Code Quality

A critical factor in the adoption of AI-powered tools into software development workflows is effectiveness in completing coding tasks.

Li et al. [34] tested GitHub Copilot, an AI-powered tool designed for software development, on problems from popular code evaluation datasets, observing that while AI-generated code is often functionally correct, it "exhibits performance regressions compared to code solutions crafted by humans." Additionally, Azeem et al. [6] concluded that while AI tools can assist in generating code, they cannot fully replace developers due to similar limitations.

Nevertheless, despite the shortcomings of code generated solely by AI, its practical use in development typically involves augmenting developer work. In this context, AI has seen strong performance. For instance, Kumar et al. [30] demonstrated "productivity improvements" and increases in "code shipment volume." Similarly, Pen et al. [40] suggested that AI tools used to aid development carry "statistically and practically significant impact on productivity."

### 2.3 Hesitancy Towards New Technologies

Inherent to any novel technologies and solutions is skepticism. In a 2017 study, Breward et al. [10] expressed that feelings toward controversial IT technologies are formulated by contextualized benefits and concerns. Zhang [57] investigates technology hesitancy from the angle of older individuals, noting that "personal characteristics... and contextual factors" are influential in formulating attitudes.

In the current context of AI, Chaturvedi and Dasgupta [13] gathered the sentiments of 20 AI experts in middle-level management positions regarding AI usage within strategic decision making (SDM). They found concepts such as "dependence, evaluation of emotions, and building trust" helpful in improving hesitancy. However, Parikh [39] notes that although AI tools can reduce development time and cut costs, concerns of accuracy and reliability persist. Rico and Öberg similarly noted opportunity for automated debugging and code optimization, but also concern with intellectual property [44].

## 3 Background

This study is motivated by the limited research on software managers' perspectives regarding AI's impact on software development. In this section, we define relevant key terms and present the research questions that guide the focus of this paper.

### 3.1 Key Terms

*3.1.1 Software Managers.* We defined software managers as professionals with significant managerial responsibilities and possessing technical knowledge of development [37], categorized as "Product manager," "Project / Program manager," or "Software development manager," and allowing for self-identification. We focused on sampling individuals currently leading a development team and possessing access to metrics representative of their team's performance.

*3.1.2 Development Metrics.* Development metrics refer to quantifiable measurements that evaluate the performance, quality, and efficiency of a development process [26]. To this end, we selected metrics accessible to software managers and define them below:

- **Number of Coding Bugs** – The count of errors identified in the software during development, testing, or after release.
- **Test Coverage Range** – The percentage of code, features, or scenarios exercised by automated or manual tests.
- **Deployment Cycle Time** – The time taken to move a code change from development through testing into production.
- **Number of Support Tickets** – The count of user-reported issues or inquiries submitted through a support system.
- **Product Building Cost** – The total expenses associated with designing, developing, and delivering the product.
- **Number of Vulnerabilities and Exposures** – The count of known security flaws in the software.

*3.1.3 Developer Characteristics.* Developer characteristics describe individual skills, talents, and habits relevant to software development [53]. Below, we list characteristics included within this study:

- **Ability to ask effective questions** – The skill of asking thoughtful questions to gain needed information.
- **Independent thinking** – The ability to analyze situations and make decisions without entirely relying on others.
- **Fast learner** – The ability to quickly understand new contexts and topics and apply them effectively.
- **Curiosity** – A genuine desire to understand how and why things work, and taking initiative to learn more.
- **Communication and collaboration** – The capacity to clearly share ideas, listen to others, and work on a team.

*3.1.4 Knowledge Work.* Knowledge work involves creating, synthesizing, and refining information in order to communicate insights effectively to others [41] [19].

## 3.2 Research Questions

Our research goal is to examine the opinions of managers when asked how AI affects the software development process. We decomposed this question into three distinct topics: developer impact [27], managerial impact [11], and job market impact [9]. Thus, within the context of the managerial perspective, we developed the following central research questions (RQs):

*3.2.1 What is the impact on developers?* As the primary drivers of code design, software developers now encounter new dynamics introduced by AI tools [27]. This evolution of workflow processes is overseen by software managers. Accordingly, we asked managers questions on how AI has shifted benchmark development metrics, and what characteristics among developers will be valued most as AI becomes more prevalent. In sum, we ask managers:

> **RQ 1:** *How have developers been affected by AI?*

*3.2.2 What is the impact on managers?* Software managers, who oversee the development process, are themselves affected by AI. Building off the works listed in Section 2.1, we investigate whether and how managers currently incorporate AI into their own practices, as well as their personal attitudes toward it. Thus, we ask:

> **RQ 2:** *How have software managers been affected by AI?*

*3.2.3 What is the impact on the job market?* The job market directly feeds the human labor needed for software development [38]. Software managers are central in this talent acquisition by defining team needs, conducting interviews, and making hiring decisions [36]. With such an involved role, we ask managers the following:

> **RQ 3:** *How has the job market been affected by AI?*

## 4 Methodology

This section describes the methodology employed within this survey, including the survey design, pilot study, participant recruitment, data collection, and subsequent analysis process.

## 4.1 Survey Design

We designed an online questionnaire using the tool Qualtrics[2]. It is located at the end of Section 4.2 as an anonymous link. This questionnaire contains four different sections. The first focuses on participant demographics. The subsequent three sections each relate to one of the three central research questions defined in Section 3.2. Respondents were prompted to complete multiple-choice, ranking, or free-response questions, formulated by the authors, each designed to help develop an answer to one of the central research questions. Additionally, our survey's content was then validated via a pilot study, which is detailed within Section 4.2 below.

With our respondents being software managers, ensuring confidentiality was key. At tech companies, there may be issues in disclosing private, company-specific practices. Thus, to protect our participants, we disabled Qualtrics' metadata collection and designed demographic questions to minimize result traceability.

Upon survey completion, participants were presented with the opportunity to redeem a $15 gift card incentive. Respondents were shown a validation code and instructed to email the code to a reward distributor, who would respond with a valid gift card number. This process was designed to prevent automated submission by malicious actors. In keeping with our efforts to maintain confidentiality, the staff member responsible for reward distribution was not an author of this paper and was restricted from any survey data. Likewise, the researchers in charge of this study and subsequent data analysis were prohibited from accessing participant contact information involved with the reward process.

## 4.2 Pilot Study

To ensure study validity, 2 software professionals provided feedback on our survey. The first professional is a technical program manager with 15+ years of experience in the industry, and the second professional has experience as a product manager, along with 11+ years of work in both academia and industry.

The feedback of these two professionals resulted in a few minor changes to the initial survey draft, with the addition of more options to a multiple-choice question on software metrics, as well as the option to provide an open-ended response to a multiple-choice question. The final version of the survey is hosted anonymously online[3], together with the survey instrument and logic[4].

## 4.3 Participant Recruitment

To recruit participants, we employed four non-probabilistic sampling methods. (1) Purposive sampling involved sending survey requests to individuals we considered likely to respond. We used the direct message feature of LinkedIn[5], a professional networking platform, and email to contact eligible professionals in our network. (2) Convenience sampling was completed at the authors' institution by inviting eligible alumni to participate. (3) Self-selection sampling was done by posting public invitations to complete the survey on various online software development forums (LinkedIn, Slack, Reddit, and DEV.to). (4) Finally, snowball sampling was done by

---

[2]https://www.qualtrics.com/
[3]https://graphhost921.github.io/survey-questions.pdf
[4]https://graphhost921.github.io/survey-flow-logic.pdf
[5]https://www.linkedin.com/

requesting respondents share the study with others who fit the sampling frame.

To verify respondents' status as software managers, the survey link was private and shared only via LinkedIn in order to verify their position. If a participant lacked a LinkedIn account or held a title different from "software manager" or a similar role, we assessed their day-to-day responsibilities. Qualification was determined by whether their duties sufficiently overlapped with the software manager role as defined in Section 3.1.1. In one case, an individual at a small startup company combined executive responsibilities and routine managerial oversight of a software development team. The excerpt below comes from another participant whose title differed from "software manager", but who nonetheless qualified.

> "I am involved in the day-to-day work with engineers, both managing them, reviewing their work, and direct collaboration on software development as well."

### 4.4 Data Collection and Analysis

*4.4.1 Dataset.* Initially, 58 software managers decided to open the survey and read the consent form. From this number, 42 completed responses were collected and utilized for analysis, providing our study with a **72.41% completion rate** (42/58).

*4.4.2 Qualitative Data Analysis.* In this study, we employed an iterative, open-coding process comprising four stages for open-ended questions, with theme development through multiple passes and discussion among the authors, and disagreement resolution through consensus. Initially, the first two authors independently coded the first third of the responses. They then met to reconcile their analyses and establish a preliminary consensus-based codebook. This codebook was then applied independently to the second third of the data, after which the authors met again to update the codes. This process was repeated for the final third of the dataset, resulting in a final codebook. This codebook is hosted online anonymously[6].

To ensure accuracy and reduce bias, an external researcher experienced in qualitative methods reviewed our final codebook. This independent auditor agreed with the codeset, confirming validity.

Inter-rater reliability was assessed using Cohen's kappa across three rounds of coding. The overall mean kappa was $\kappa = .78$ (SD = .15), indicating substantial agreement [31].

## 5 Results and Discussion

This section presents the survey results, analysis, and discussions. First, we detail the demographics of our dataset. Then, we examine the answers provided by software managers in the following sections, representing each research question posed in Section 3.2.

### 5.1 Demographics

Of the 42 software managers who responded, 38 identified as men ($38/42 \approx 90.48\%$), 3 as women ($3/42 \approx 7.14\%$), and 1 as non-binary ($1/42 \approx 2.38\%$). While this demographic imbalance is a study limitation, it may also indicate a broader disparity in the software manager position and the industry as a whole. Further discussion on this topic can be found in Section 6.2.

___
[6]https://graphhost921.github.io/codebook.pdf

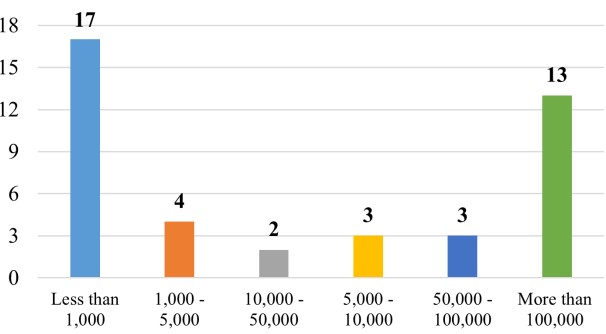

**Figure 1: Company Size Distribution**

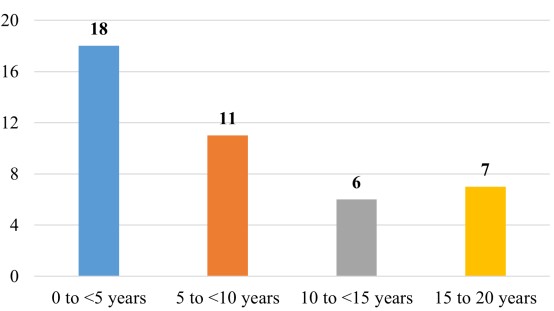

**Figure 2: Years of Experience Distribution**

When asked about age range, 8 managers reported being between the ages of 25 - 34, 21 between 35 - 44, and 13 between 45 - 54.

Responding managers primarily work at companies with fewer than 1,000 employees ($17/42 \approx 40.48\%$) and companies with more than 100,000 employees ($13/42 \approx 30.95\%$). This wide range provides a broad window into how software managers at both small and large companies view AI's effect on the development process. This data is displayed in Figure 1.

Software managers then inputted the number of years and months of experience they possessed. Within this study, software managers possessed an average experience of 7.54 years, a median of 5.75 years, a maximum of 20 years, and a minimum of 6 months. Figure 2 presents this distribution within 5-year ranges.

When asked about role title, 31 managers selected "Software development manager", 4 selected "Project / program manager", 0 selected "Product manager", and 7 selected "Other - please specify". Table 1 lists the roles entered by managers who chose to manually specify their role title. Some of these titles are not obviously associated with the software manager role. However, these participants' responses were counted as their responsibilities fit under the definition of a "software manager" as defined in Section 3.1.1. Further elaboration for participant selection criteria can be found in Section 4.3.

Notably, our research team selected not to stratify results along demographic lines due to the unevenness of the data and limited number of responses. This decision is elaborated on in Section 6.2.

| Job Title | Count |
|---|---|
| CTO: | 1 |
| DevOps Manager: | 1 |
| Engineering Director: | 1 |
| Exec: | 1 |
| Software Engineering Manager: | 2 |
| Software QA Manager: | 1 |

**Table 1: Other Job Titles**

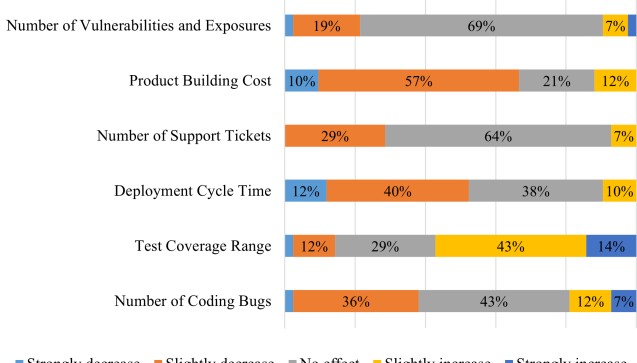

**Figure 3: Perceived Impact of AI on Development Metrics**

## 5.2 Impact on Developers

The questions in this section investigate how managers view AI's impact on the developer within the software development process.

We first asked managers how much they or other leadership encourage their developers and teams to use AI. 11 managers selected "A great deal," 16 chose "A lot," 12 answered "A moderate amount," 2 said "A little," and 1 indicated "None at all." We classify "A great deal" and "A lot" as strong encouragement, so more than half of software managers ($(11 + 16)/42 \approx 64.29\%$) provide strong encouragement for AI usage. Within our sample, **a majority of software managers encourage their teams to use AI in development.**

Following this, respondents then rated how they believed certain software metrics would be affected by the introduction of AI into the development process. The results from this question are listed in Figure 3. Within this portion of analysis, we highlight two insights:

- "Product building cost" was primarily perceived by managers to decrease, with 57% perceiving a slight decrease and 10% perceiving a strong decrease
- "Deployment cycle time" was perceived to slightly decrease by 40% of managers, and strongly decrease by 12%
- "Test coverage range" received a majority of perceived increase, with 43% perceiving a slight increase, and 14% perceiving a strong increase

Software managers were then prompted to rank five characteristics within developers that would be most valued with the rise of AI, assigning a number from 1 (most important) to 5 (least important). Across all responses, the average ranking for each characteristic

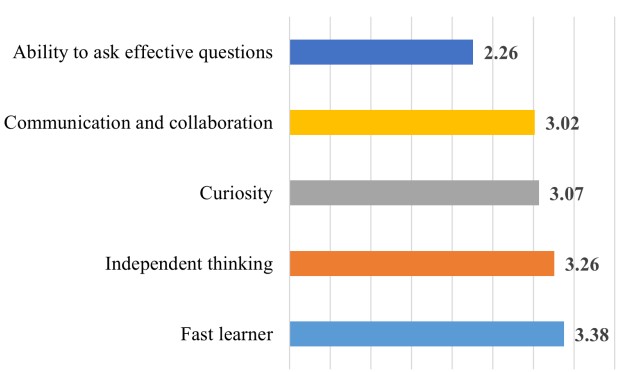

**Figure 4: Ranking of Developer Characteristics**

was calculated and are displayed in Figure 4. Given the responses, **the highest-ranked skill was the "Ability to ask effective questions"**. Software managers in our sample clearly ranked this skill as the most valuable. This is understandable, as effective use of AI tools depends on framing precise, meaningful, and contextually relevant prompts or questions [29]. Without this ability, even advanced AI models may deliver poor results. Managers may therefore value developers who can question and critically interpret the material AI creates. Interestingly, "Independent thinking" and being a "Fast learner" ranked at the bottom. At least among our respondents, this result possibly reflects a shift in the traditional prioritization of independent thinking and fast learning in software engineering. AI tools may alleviate the burden on developers to quickly master unfamiliar concepts. Now, the emphasis may lie in knowing what to ask AI and how to question its results, rather than individual coding skill.

Lastly, we asked software managers to freely respond to: "In what areas do you see the software development process evolving with the rise of AI?" Throughout the responses, the **application of AI for testing purposes was clearly a prevalent sentiment**. Many responding managers believe that AI can reduce test costs by generating and implementing test code, as well as increasing the rate of prototyping, proof of concepts, and building minimum viable products (MVP). This sentiment aligns with our previous result in Figure 3, where a majority of managers viewed AI as leading to an increase in software test coverage range. One manager provided example applications of AI for testing purposes:

> "I foresee great evolutions across... the software testing spectrum (e.g., fully automatic integration tests, end to end tests with "natural language" playbooks, etc.)"

Beyond usage for testing, two other ideas stood out, albeit to a lesser extent: "documentation" and "code generation." Managers believe that AI can be valuable in creating documentation for the team, fostering collaboration. Managers also believe that AI can help generate code and assist developers in their work. In relation to software developers, one participant noted:

> "I truly believe that AI can be used effectively if the company utilizes it as a force multiplier and not a way to reduce headcount."

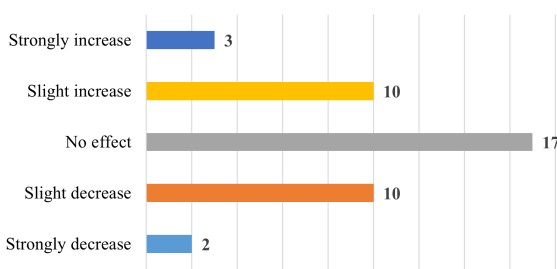

**Figure 5: Workload Change**

## 5.3 Impact to Managers

This section examines how managers view themselves and their work to be affected by AI as the technology grows in prominence.

We initially asked software managers if they were involved in their company's hiring process. The 40 managers who answered "Yes" were presented with two follow-up questions. First, the managers were asked if they used AI to filter job applications. Interestingly, a majority of participating managers (30/40 = 75%) don't utilize AI to filter applications. While this result seemingly misaligns with the idea that AI filtering tools are finding increasing usage within companies' hiring practices, an approach documented within previous works [3] [43], our result may simply reflect the practices of the particular managers and companies surveyed. Further research should explore the extent of the presence of AI tools for hiring.

We then asked software managers if they listed AI as a prerequisite for the roles for which they were hiring, with a majority 26 responding "No" (26/40 = 65%). While this may indicate that AI is not a prevalent skill across software development roles, this may change. A point of future research may be to examine this percentage over time.

Next, we asked respondents if their workload had changed with the introduction of AI. Visualized in Figure 5, most managers selected "No effect" (17). Very few chose "Strongly increase" (3) or "Strongly decrease" (2). Notably, this distribution of responses is symmetrical, with no clear upward or downward trend. Perhaps, the cause for this mixed response is AI hesitancy, limited impact on duties, or benefits offset by potential costs from AI-augmented work [22]. What is most likely, however, is that software managers will always face a consistently high workload from their employers [37] [51], regardless of whatever tools exist to optimize their work.

Software managers were then asked: "How do you use AI in your own personal work, if at all?" Throughout their responses, **managers reported primarily using AI for knowledge work tasks**, with three distinct codes falling under this category. Listed from higher to lower frequency, the codes we developed are:

(1) **Document creation** - This involves drafting documents, notes, emails, reports, and tickets from scratch by using AI. With prompting, these tools can generate text rapidly [12], helping managers circumvent the "writer's block" [2], reducing the time spent to start the writing process.

(2) **Information synthesis** - Managers summarize documentation, emails, meeting notes, and customer tickets using AI.

Managers also described using AI as an enhanced search tool to gather information quickly.

(3) **Communication refinement** - Parallel to document creation, managers use AI tools for grammar checking, tone adjustment, and clarity, especially when communicating with stakeholders, clients, or upper management [32] [42].

We then asked software managers, "What are your biggest ethical concerns with the widespread adoption of AI in the software development process, if any?" Software managers **most frequently cited privacy as an issue**. Responses reflected concerns that AI may mishandle sensitive information, as some AI services collect private user data [17] [20]. One manager bluntly wrote:

> " [AI is] not as transparent as they say. Fakes of everything, trust issues... Personal and corporate damage."

Managers also flagged security risks, especially organizational data exposure. Another cited concern was responsibility and ownership over AI-generated material. Specifically, managers questioned who would be held accountable if AI-generated code caused harm in production: the developer using the tool, the authorizing manager, or the AI provider. The lack of established legal clarity intensifies these ethical concerns [47] [46]. One manager noted:

> "[AI is] an inevitable tool that will change the development of software; therefore the importance of proper guidelines and standards cannot be understated."

Similarly, managers noted the lack of transparency and explainability in AI tools, which further impedes accountability [56]. Several respondents also expressed concern over the potential dehumanization of operations. One manager strongly commented:

> "Agentic AI is... an attempt to replace large swaths of subject matter without the safeguards of organizational structures and agendas to counterbalance."

Software managers identified additional ethical concerns regarding AI. Some managers highlighted the risk of intellectual property theft, noting that AI models may incorporate proprietary or confidential data to generate their output [49]. Other managers pointed out inherent biases that may persist within AI systems, originating from flawed training data, which can reinforce social inequalities and lead to unfair decision-making. AI hallucinations [45], a phenomenon in which models generate inaccurate or fabricated information that misleads users, was also flagged as a risk.

Our final question for this section asked managers: "Besides ethical concerns, is there anything else you want to add about your attitude toward AI in the future of the software industry?" As this question was optional, 26 managers responded. They most often cited long-term risks caused by excessive dependence on AI and **the decline of higher-order abilities**, including critical thinking, communication, and information synthesis. Notably, this concern distinctly echoes earlier answers on how managers themselves use AI. Some managers predict that reliance on these tools, for coding or knowledge work, will subsequently deteriorate skill in these areas. This concern extends to new, inexperienced developers:

> "[I worry about] loss of basic coding and debugging skills... [and] basic [software] architecture skills [among] newer developers [who are] dependent on AI"

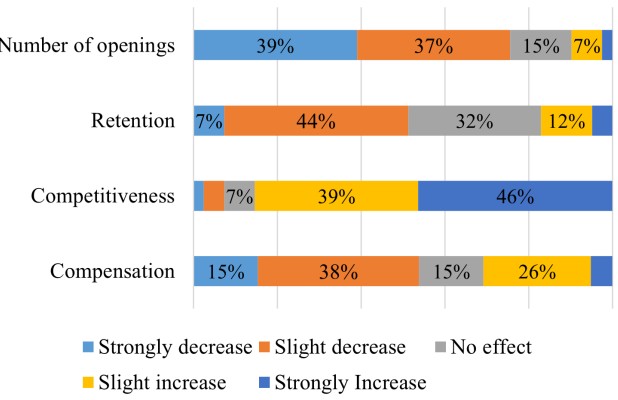

**Figure 6: Perceived Impact of AI on the Job Market**

In addition to dependence, managers mentioned AI output accuracy problems, a broad area encompassing both factual mistakes as well as ethical dangers tied to inherently biased outputs. Taken altogether, these concerns show that software managers view AI as both a helpful tool and a potential long-term liability. This nuance is embodied by one manager's response:

> *"Many of us assume that if [they're] not spending time deep diving into fundamentals... the next wave of software developers will be less capable. However, I'm very curious if that will hold true because I do wonder if... [leveraging] AI in terms of development will compensate."*

### 5.4 Impact to Job Market

This final section of analysis focuses on how managers perceive the software development job market to be affected by AI.

First, software managers were prompted to rate how they believed certain aspects of the job market would change as a result of AI. Below, we discuss a few notable details as displayed in Figure 6.

- "Number of openings" was generally perceived by managers to decrease, with 37% perceiving a slight decrease and an even larger 39% perceiving a strong decrease
- "Competitiveness" was overwhelmingly viewed by managers to increase, with 39% perceiving a slight increase and 46% perceiving a strong increase

Our final question asked: "How else do you think the introduction of AI will affect the software development market?" Software managers frequently **emphasized potential job losses** in particular, reflecting the result of the previous question. According to respondents, such displacement can manifest in various forms, from the elimination of entry and junior level roles to the general streamlining of development teams. They believe that AI plays a major role in this shift, as while it empowers individual coding ability, companies may consequently consolidate responsibilities that previously required multiple entry-level engineers [15]. However, in contrast, a few managers shared the sentiment that AI will enable individuals with non-technical backgrounds to develop software, perhaps creating new opportunities. One manager articulated:

> *"Short term, we are already noticing job reductions, and a fierce competition in the data-science domain. Long term, I think [that] AI will eventually stabilize... and we will find new forms of collaboration, productivity and satisfaction in software development."*

In addition to potential job displacement, managers highlighted a shift in the expertise required in the evolving software development labor market. They note AI skills, such as prompt engineering, are increasingly regarded as an essential qualification. Further responses from managers include potential compensation reduction, heightened selectivity in hiring practices, and concerns regarding an economic bubble associated with AI technologies [23].

## 6 Threats to Validity

This section details threats to the validity of our study, and the steps taken to mitigate their influence. We grouped these possible issues under two categories: construct and external.

### 6.1 Construct Threats to Validity

Construct threats to validity arise when data collection methods misrepresent a study's focus. One potential issue was a misalignment between research intentions and survey questions. Thus, we completed a pilot study, implementing feedback from two software managers. This process is detailed in Section 4.2. Additionally, all respondents were presented with a consent form outlining research goals and contact information for questions. Finally, we reviewed responses manually to ensure alignment with our study's goals.

### 6.2 External Threats to Validity

External threats to validity refer to issues that limit the generalizability of the results to the general population. With the managerial perspective being niche and hard-to-reach, addressing this concern was key.

(1) Ensuring the collection of a sufficient number of responses was necessary for academic value. Our outreach yielded 42 completed responses from software managers. This result adds both useful data on AI's influence in software development and highlights the challenge of collecting managerial perspectives. Busy schedules and steep wages make survey participation rare, even with an incentive. To strengthen confidence in our results, we plan to conduct a future study with a larger dataset and supplement it with in-person interviews for deeper insights.

(2) Generalizing findings across a multifaceted industry is inherently difficult, as the managerial experience can differ widely by company size, environment, and background. We initially included demographic questions to enable stratification and assess representativeness. However, the resulting dataset was highly unbalanced across several demographics lines, limiting meaningful stratified analysis. For example, when asked for role title, most respondents selected "Software development manager" ($31/42 \approx 73.81\%$). The next largest selected role of "Project / program manager" was far smaller ($4/42 \approx 9.52\%$). In another case, most participants identified as men ($38/42 \approx 90.48\%$), possibly indicating an underrepresentation of women and non-binary managers. While these groups are broadly underrepresented in computing roles, this imbalance remains a study limitation. Ultimately, our team decided to

forgo stratification altogether and instead treat software managers as a single analytical group, even as the sample imbalance did not occur across all demographics. This choice reflects the exploratory nature of the study, rather than a claim of response homogeneity. Our goal is to provide an initial window into the perspective of software managers and motivate future research that more deliberately examines demographic variation within this population.

(3) Our sampling strategy introduces potential selection and sampling biases. By distributing our survey through online platforms, as listed in Section 4.3, we may have disproportionately reached software managers who are more active in online communities. This trait could potentially correlate with other characteristics, such as managers' age, gender, years of experience, or company. As our research was also voluntary, the managers who responded may also possibly hold stronger opinions about AI, thereby limiting the representativeness of the sample. Similarly, our use of convenience sampling through alumni networks may have led to an overrepresentation of managers affiliated with specific institutions, geographic regions, or company types. Despite these limitations, the study is a valuable exploratory snapshot of software managers' sentiment toward AI in the software development process, especially given the difficulty of accessing this population. We recommend that future work should aim to mitigate these biases through more systematic sampling strategies across industries, company sizes, or experience levels.

## 7 Broader Impact

AI has emerged as a powerful tool with the potential to improve various aspects of the software development process. Yet, as evidenced throughout our study, its adoption remains contested, and software managers display a highly nuanced stance. In this section, we present the broader impact of our paper, including recommendations to software professionals, the ethical implications, as well as a roadmap for future endeavors.

### 7.1 Guidance for Software Professionals

The software industry is a highly dynamic field that expands or contracts with the advent of each new innovation. Maintaining professional relevance in this evolving field requires continuous adaptation, especially as software managers expect AI to increase competitiveness in the job market. Thus, we recommend the following:

*7.1.1 For software developers:* Many current managers encourage their teams to leverage AI in their work, valuing those who possess strong questioning skills, potentially related to the ability to prompt and interpret the output of AI. This sentiment likely signals that AI will exist, at least partially, as a core job expectation.

> **Software developers:** It will prove useful to possess AI competencies within software development, including the ability to demonstrate skill in prompt engineering and critical evaluation of AI outputs.

*7.1.2 For software managers:* Our findings indicate that many peers particularly utilize AI for knowledge work, such as document creation, information synthesis, and communication refinement. Additionally, they perceive that their team's test coverage range will increase as a result of AI, potentially cutting costs through test code generation and implementation.

> **Software managers:** AI tools can be beneficial in knowledge work. Additionally, exploring the integration of AI tools into the testing process of the team's development workflow may reap benefits.

### 7.2 Addressing Ethical Concerns

The successful integration of AI tools into the software development workflow requires a spirited effort to address the ethical concerns held by software managers. Below, we list the most frequently voiced issues and approaches to address them.

Managers expressed concerns that AI models may collect private data or mishandle sensitive information. We thus recommend:

> **Privacy and security:** AI tools used for software development must be audited, ensuring adherence to data minimization principles to protect sensitive data and compliance with applicable privacy regulations.

Respondents raised concerns about who is responsible for the outcomes of AI tools, especially within situations involving negative consequences. Accordingly, we suggest:

> **Ownership and accountability of AI-generated material:** Organizations should develop guidelines and accountability chains that clearly define ownership of AI-driven decisions or code.

Software managers highlighted a concern that excessive automation may reduce meaningful human oversight in software development. We therefore advise:

> **Dehumanization of operations:** Organizations should ensure active human involvement within the software development process, especially within ethically sensitive contexts.

Managers presented concern about the risk of inherent bias in AI systems, possibly due to flawed training methodologies. We urge:

> **Bias:** The creation of AI tools must utilize diverse datasets and input from individuals with varied backgrounds. Building diverse development teams can help identify and reduce inequities in AI tools.

Common to each concern is the need for clear policy. We strongly advocate that the IT sector must develop enforceable policies to ensure the responsible usage of AI tools. Whether driven by industry leaders, policymakers, or standards bodies, the concerns raised by software managers points to the urgent need for robust and coordinated AI governance.

### 7.3 Roadmap for Future Endeavors

The insights gathered in our paper reveal a complex relationship between the enormous potential and multifaceted challenges of AI. Our findings call for further analysis of the software manager's perspective. Below, we offer several promising research directions.

(1) Managers' concerns regarding skill erosion are consistent with current literature [21] [52]. In the field of software development, it entails the over-reliance on AI tools, leading to a decline in basic programming, debugging, and design skills in developers. Future research should pursue this topic, possibly a comparative analysis between junior and senior developers.

(2) Given the rise of AI, managers rated "The ability to ask effective questions" as the most valued characteristic in developers. This emphasis likely reflects a developer's ability to prompt AI models and evaluate their outputs. We suggest that future research should develop ways to foster and assess effective human–AI collaboration [25] [33].

(3) As discussed earlier in Section 6.2, analysis of software manager's sentiments along demographic lines is important. Our study provides an initial look into the general managerial perspective, and future research should take a more targeted approach and compare managers' sentiments toward AI along lines of experience, gender, or company size.

## 8    Conclusion and Future Work

Our study provides an early examination of how software managers perceive the impact of AI on software development. Through a survey of 42 managers, we found that responding managers strongly encourage the use of AI by developers, particularly for documentation and code testing. Within our sample, managers use AI for productivity in their daily work, while also maintaining ethical concerns including privacy and security, ownership and accountability of AI, dehumanization of operations, and bias. In addition, managers anticipate a contraction in the job market, with fewer openings and increased competitiveness, especially for entry-level roles. These findings highlight the nuanced and dualistic nature of AI's role in software engineering: it is both an elixir that enhances productivity and innovation and a source of new challenges that require careful management. As AI continues to transform software development, the manager's perspective will remain critical in understanding and guiding this transition. Ensuring that AI serves as a sustainable and empowering force is integral to the software engineering community.

We plan to conduct a future study that samples a larger dataset and supplements it with in-person interviews for further insights. We are also particularly interested in the following two research directions. First, based on the various ethical concerns expressed by managers, we are interested in the investigation of the impact of organizational and regulatory policies in shaping the ethical and effective use of AI in software engineering. Second, we also observe the need to develop frameworks, tools, and training programs to help managers and developers mitigate AI-related risks, such as over-reliance, bias, and security vulnerabilities, while maximizing the benefits brought by AI.

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
