# OpenReview forum: "Is Artificial Intelligence an Elixir to the Software Engineering Community? An Empirical Study Among Managers"
_ACM.org/AIWare/2026/Conference — AIware 2026_

### Official Review · Reviewer_HbwS · 2026-02-21

**Rating:** 4
**Confidence:** 3

**Review:**

Pros:

Quality: Not being an expert in human subject study, I find this study rigorous and useful.

Clarity: The presentation of the study is structured and clear.

Originality and significance of this work: To my best knowledge, this work is the first to survey managers for their view on AI usage in SE.

Cons:

1. Missed related work:

Rico, Sergio, and Lena-Maria Öberg. "Challenges and Opportunities for Generative AI in Software Engineering: A Managerial View." In Proceedings of the 33rd ACM International Conference on the Foundations of Software Engineering, pp. 1338-1344. 2025.

Assalaarachchi, Lakshana Iruni, Zainab Masood, Rashina Hoda, and John Grundy. "Generative AI for Software Project Management: Insights from a Review of Software Practitioner Literature." IEEE Software (2025).

**Summary:**

This paper investigated the managers' view on the application of AI to software development. An online survey with both quantitative and qualitative questions was conducted to collect inputs from 42 managers. The survey results suggested that (1) AI usage is beneficial for both software developers and managers; (2) ethical and privacy concerns were raised along with the concerns for "“deskilling"; (3) AI usage has impacted the job market and reduced developer jobs already.

---

> ### Author Response · Authors · 2026-03-21
> **Acknowledgment of Suggested References**
>
> We thank the reviewer for providing recent related work.
> We will incorporate and discuss these references in the camera-ready version if the paper is accepted.

---

### Official Review · Reviewer_KNqi · 2026-03-10

**Rating:** 3
**Confidence:** 5

**Review:**

Method and Rigor. The paper reports a survey study with software managers about the impact of AI tools on software development. The overall approach makes sense for the research questions, since the goal is to understand managers’ perceptions and experiences. The authors describe how participants were recruited through LinkedIn, alumni networks, online communities, and snowball sampling. They also report that the questionnaire was implemented online and mention a pilot with two professionals before running the full survey.

However, the paper does not really explain how the questionnaire was designed in the first place. It is not clear to me how the questions were derived from the research questions, whether they were inspired by previous surveys, or how the constructs being measured were defined. This makes it difficult to evaluate the validity of the instrument. Another point that needs clarification is the analysis process. Some of the survey responses appear to be open-ended and the paper mentions coding them into themes. However, the paper should explain the coding process more clearly. For example, it would help to know how the codes were developed, whether coding was done by multiple researchers, and how disagreements were resolved. Without these details, it is hard to assess the rigor of the analysis.

-----

Goal, Results, and Relevance. The main goal of the paper is to understand how managers perceive the influence of AI tools on developers, management activities, and the broader job market. This perspective is interesting because most discussions about AI in software engineering tend to focus on developers rather than people managing development teams. The results provide some useful observations about how managers think AI may affect productivity, development practices, and future hiring. However, the findings mostly describe opinions from the participants rather than establishing broader patterns. Since the sample is relatively small and comes from convenience-style recruitment, the results should probably be interpreted as exploratory rather than representative of the wider software engineering industry. Still, the study raises several points that could motivate future work, especially larger surveys or studies that combine perspectives from both managers and developers.

-----

Verifiability and Transparency. The paper provides a general description of how participants were recruited and how many responses were collected. It also reports the completion rate and includes demographic information about the respondents. However, a few details that would improve transparency are missing. In particular, there is no discussion of potential response bias related to the sampling process. Also, since qualitative coding appears to be involved, more detail about the coding procedure would make the analysis easier to understand and replicate.

-----

Presentation. Overall, the paper is easy to read, and the structure is clear. The research questions are straightforward, and the results are presented in an organized way.

-----

Overall Perception.
The paper looks at an interesting and timely topic. The survey provides some initial insights into how managers perceive changes in productivity, workflows, and the future workforce. At the same time, the empirical part of the paper feels somewhat preliminary. The questionnaire design is not fully explained, and the qualitative analysis would benefit from a clearer description of the coding process. With a bit more detail on the survey design and data analysis, the study would feel much more rigorous.

**Summary:**

This paper reports a survey of 42 software managers to understand how they perceive the impact of AI tools on developers, management roles, and the software development labor market.

---

> ### Author Response · Authors · 2026-03-21
> **Response to Comments on Methodology, Rigor, and Transparency**
>
> We thank the reviewer for the thoughtful and constructive feedback.
>
> [Method and rigor] We agree that the design of the questionnaire would benefit from a clearer description. The questions were derived to align with our core research themes, informed by both prior literature [23] [32] and real-world industry experience from one of our co-authors. In addition, based on our limited knowledge, no survey-based empirical study has been conducted to understand the impact of AI perceived by software managers.
>
> In the camera-ready version, we will revise the paper to more explicitly describe how the research questions informed the questionnaire design, including the rationale behind key question groups and how constructs were derived.
>
> [Exploratory nature and generalizability] We agree that, given the sample size and recruitment approach, the results should be framed more as exploratory rather than representative. We will make this positioning explicit throughout the paper and ensure that claims are appropriately qualified in our camera-ready version if the paper is accepted.
>
> [Verifiability and transparency] We appreciate the request for more detail and agree that the current description is too brief. Thus, we will specify that our analysis involved iterative coding of open-ended responses, with theme development through multiple passes and discussion among the authors, and disagreement resolution through consensus.
>
> We appreciate the review and will implement suggestions to improve the rigor and transparency of the paper in our camera-ready version if the paper is accepted.

---

### Official Review · Reviewer_Q17m · 2026-03-13

**Rating:** 4
**Confidence:** 3

**Review:**

### Pros

- Very timely and relevant perspective for AIware
- The question design is well considered -- both closed-ended and open-ended questions
- Practical and actionable implications for developers and managers

---

### Cons (kindly see my detailed comments)
- Sample size is small and somewhat skewed
- Missing inter-rater reliability metrics (e.g., Cohen's kappa) for the qualitative analysis
- Several survey questions can have ambiguity
---

### Detailed Comments

Overall, this is an interesting paper that surveys software managers and surfaces many valuable insights and suggestions for software professionals, along with a useful roadmap for future research. That said, I have a few concerns as follows.

One concern is the mismatch between the strength of the claims and the evidence base like sample size and distribution. n=42 responses is a reasonable amount and I believe it is already quite good -- given the exploratory nature of the study, but some parts of the paper present findings with an overly strong level of confidence. For instance, the observation that 75% of managers do not use AI to filter applications is presented as a notable finding that "contradicts" prior literature, but with only 40 respondents answering this question, such a comparison would carry weight or not? Maybe the authors should consider to be more careful in qualifying their claims throughout, like the phrases "our results suggest" or "among our respondents" would be more appropriate than definitive statements. Also the decision to forgo stratification, while understandable given the imbalanced demographics (90% male, 74% holding the same job title), further narrows the analytical contribution of the quantitative data.

The qualitative analysis somewhat falls short of current standards in empirical SE. The iterative coding process is good, but without Cohen's kappa or Krippendorff's alpha values at each reconciliation stage. Also, the external auditor's role is described in a single sentence with no detail on the evaluation criteria or process, which should be elaborated. If the auditor merely confirmed that the codes "made sense" rather than independently applying them to a data subset, this does not constitute meaningful validation. The authors could either compute and report inter-rater reliability metrics or, at minimum, provide a substantially more detailed account of the auditor's evaluation procedure and scope.

The survey design also suffers from ambiguity in several places. For the metrics question, it is unclear whether managers are reporting changes they have already observed or speculating about anticipated future effects, and no baseline or timeframe is specified. For example, I expected a manager at a company that adopted AI tools two years ago and one that has not yet adopted them would likely interpret this question very differently. The forced-ranking design for developer characteristics also requires respondents to impose a total ordering on five items, which conflates items that may be perceived as nearly equal in importance and does not capture the magnitude of differences between them. In future iterations, the authors may need to reframe these questions and present the empirical findings in a more nuanced way.

**Summary:**

This paper presents an empirical survey study examining software managers' perspectives on how AI tools affect software development. Through an online questionnaire completed by 42 software managers, the study investigates three dimensions: (1) AI's impact on developers, (2) AI's impact on managers themselves, and (3) AI's impact on the software development job market. The findings suggest that managers largely encourage AI adoption, perceive AI as valuable for testing and knowledge work, and apply it in their own roles for tasks such as document creation and information synthesis. Meanwhile, they raise ethical concerns around privacy, accountability, transparency, and over-reliance, and anticipate job market contraction, particularly for entry-level roles. The paper concludes with guidance for software professionals and a roadmap for future research.

---

> ### Author Response · Authors · 2026-03-21
> **Response to Comments on Claims, Analysis Rigor, and Survey Design**
>
> We thank the reviewer for the thoughtful and constructive feedback.
>
> [Strength of claims vs evidence] We agree that, given the exploratory nature of the study and sample size, some statements would benefit from more careful qualification. Our intent was not to make broad generalizations, but to highlight trends observed among our respondents. We will revise the wording to include phrases such as “our results suggest” or “among our respondents”, softening claims where appropriate in the camera-ready version.
>
> [Qualitative analysis rigor] We thank the reviewer for highlighting this. Our process involved iterative coding with reconciliation and external review; however, we agree that the description lacks sufficient detail.
>
> In our revised version, we will add the inter-rater reliability metrics Cohen’s kappa. We will also expand the methodology section to more clearly describe the coding procedure, reconciliation steps, and the role of the external auditor.
>
> [Survey design clarity] We agree that some questions could be interpreted differently. We mentioned this concern in the section of Threats to Validity.
>
> Regarding the forced-ranking design, we acknowledge that it does not capture magnitude differences and may conflate closely ranked items. We will explicitly note this limitation in the camera-ready version. If space is allowed, we will also refine the presentation of results surrounding this note.
>
> Overall, we appreciate these suggestions and will incorporate them to improve the clarity, rigor, and positioning of the paper.

---

### Official Review · Reviewer_bYk1 · 2026-03-13

**Rating:** 3
**Confidence:** 5

**Review:**

## Key Positives

- Addresses a genuine gap in the literature on AI adoption from the managerial perspective.
- Explores a rich and timely topic with strong follow-up potential.
- The experimental qualitative methods seem appropriate

## Key Negatives

- Lack of focus in research questions and survey design.
- Outdated related work that does not reflect the current state of AI capabilities.
- Superficial analysis of results with limited use of qualitative evidence.

My main concern with this paper is that it tries to do too much without going deep enough in any single direction. The fact that "AI" is such a broad term and covers so much ground further complicates things. Most importantly, the RQs and particularly the individual survey questions lack a proper motivation. What insights are the authors trying to get and how do those questions contribute towards that goal? The related work section relies heavily on studies from the early LLM era (e.g., including benchmarks and evaluations conducted with GPT-3). These no longer represent the capabilities of frontier models, which ends up weakening the paper's overall framing and motivation. Finally, some questions are not analyzed with enough rigour and could benefit from more quotes showing examples of the managers' specific points of view. A tighter scope, updated references, and deeper/more careful qualitative analysis with additional representative quotes would substantially strengthen the contribution. Having said that, considering the lack of studies from the managerial perspective, the study still has value as an initial exploratory contribution.

## Detailed Comments

1. [Lack of focus] The online questionnaire covers a very wide range of topics, and many questions feel either too broad or too narrow. The study would have benefited from a more focused survey aligned with fewer, sharper research questions. I would be valuable to know their perspective (perhaps with more open ended questions) on the many other critical elements of current development workflows and interactions. For instance, how should teams deal with code review workload now that code can be produced much faster (more PRs, e.g., done by agents when developers are sleeping, etc)?

2. As I mentioned before, the motivation behind the questions and/or their rationale is not very clear. For example, asking managers a predictive question about how the job market will change feels too speculative to me. I believe it would have been more insightful to ask what changes they are already perceiving right now as it betters reflects their perception of the status quo.

3. [Lack of focus] "The impact of AI on software developers" is too broad and can mean many things, such as: how does it impact their workflows, how does it impact their team interactions, how does it impact their productivity (or perceived productivity), etc. So this feels too big of a question that hasn't been broken down properly or addressed in enough detail in the survey.

4. [Lack of focus] "We then asked software managers if they listed AI as a prerequisite for the roles for which they were hiring." What exactly does that mean? Knowing how to use AI? Knowing how to use coding agents? Being simply well-versed/knowledgable of AI? What kind of AI? The unexpected result you observe (26 responding 'no') might be just because the question was imprecise.

5. [Related work] The paper frames its scope as the "influence of AI on software development" but does not cite recent relevant work (e.g., from Thomas Zimmermann@Microsoft). There's also a growing literature on coding agents that should probably be acknowledged (e.g., check https://arxiv.org/pdf/2507.15003). Consider incorporating these to strengthen the motivation and better situate the study. This is particularly important in this area where new updates occur *very* fast. There's also plenty of physical space available in the paper for additional references.

6. [Related work] The claim that LeetCode problems require human-like reasoning beyond current AI capabilities is outdated. Frontier LLMs now perform very well on competitive programming benchmarks, including LeetCode. This section should be updated with more recent references. The study about code performance with ChatGPT is also outdated, since it used GPT-3.

7. [Analysis] "within this portion of analysis, we highlight two insights." A more nuanced/detailed analysis should be performed. For instance, it is interesting to see most respondents thinking that deployment cycle decreased.

8.  [Analysis] The finding about the "application of AI for testing purposes" would be strengthened by including a few representative quotes. Similarly, if privacy is the most frequently cited concern among managers, the paper should include a few representative quotes to better illustrate the specific privacy worries. Overall, these direct quotes help the reader understand the details and reasoning behind some of the key conclusions you're making.

9.  [Analysis] "Notably, this distribution of responses is symmetrical, suggesting that software managers' workloads are largely unaffected by AI, with no clear upward or downward trend." -> This analysis is too superficial. Figure 5 shows a normal distribution. Roughly speaking, 1/3 think these's increse, another third+ thinks there's no effect, and another third thinks there's a decrease. So their perception is actually all over the place and not exactly negative ("AI hesitancy" as your say in the text).

10. [Analysis/Conclusion] "Software developers: It will prove useful to possess AI competencies within software development, including the ability to demonstrate skill in prompt engineering and critical evaluation of AI outputs" -> This sounds quite obvious given the state of practice. Consider framing this finding more carefully or connecting it to less intuitive implications.

13. I believe that the "effective human-AI collaboration" is indeed a big deal and could be expanded with more concrete and interesting directions. For example, how should SE undergraduate courses teach this skill? should more focus be given to requirements engineering, writing clarity, and critical thinking than coding skills?


## Other comments

1. "Large language models such as ChatGPT" -> ChatGPT is typically associated with the application/product (the chat interface) rather than being a model per se.

2. The subsection title "AI's Coding Performance" can be misleading, since readers might interpret it as actual performance/efficiency (as in Software Performance Engineering). Maybe change use the term "code quality".

3. The abstract simply mentions surveying "companies". It would be helpful to briefly clarify the demographics to help readers better understand the surveyed participants.

4. This is a confusing sentence: "they recommend implementing a mechanism to alleviate managers' personal concerns about the tool, placing critical consideration on the "dark side" and hidden biases of AI." In particular, what tool? AI?

5. (Abstract) The term "knowledge work" is used without clear definition.

**Summary:**

This paper presents a qualitative study on the impact of AI on software engineering from the perspective of software managers. The authors designed a survey covering three key topics: the impact of AI on software developers (Q1), the impact of AI on managers (Q2), and the impact of AI on the job market (Q3). The survey was answered by 42 managers with varying years of experience and working on companies of different sizes. Regarding Q1, the authors show that (i) the majority of managers encourage their teams to use AI in development, (ii) managers ranked "ability to ask effective questions" as the highest valued developer characteristic/skill, and (iii) that managers see the application of AI for testing purposes as the key area where software development is evolving due to the rise of AI. On Q2, the authors highlight that (i) managers primarily use AI for document creation, information synthesis, and communication refinement and (ii) managers most frequently cited privacy as a critical ethical concern. Finally, regarding Q3, the authors show that (i) managers believ4e that the number of job openings will decrease and competitiveness will increase and (ii) managers are concerned with potential job losses. Future research directions are outlined.

---

> ### Author Response · Authors · 2026-03-21
> **Response to Comments on Scope, Framing, and Related Work**
>
> We thank the reviewer for the detailed and constructive feedback. We address the main concerns below. Regarding the detailed comments section of the review:
>
> [1, 3, 4 Lack of focus] We agree that the space of “AI’s impact on software development”, even when asked specifically to software managers, is broad and that the different facets of this topic, such as impact on software developers or job prerequisites, deserve further investigation.
>
> In our paper, we sought to provide an initial look into the managerial perspective across multiple dimensions, rather than dive into a single aspect. We will clarify this position within our camera-ready version.
>
> Meanwhile, we will also add this to the Threats to Validity section in the camera-ready version if the paper is accepted, acknowledging the focus is specified in our paper, expecting to inspire future studies on similar topics.
>
> [2] We thank the reviewer for this suggestion. Our intent with predictive questions was to capture forward-looking perceptions from managers. However, we agree that framing this question within a present context is necessary to understand how managers perceive the status quo regarding AI. Therefore, in the camera-ready version, we will revise the tense of questions.
>
> [5, 6 Related Work] We truly appreciate this feedback! We agree that the related works section can be strengthened with recent work, especially the growing literature on coding agents, and we will review and incorporate this material into the paper in our camera-ready version (if accepted) accordingly. We also acknowledge that some claims, such as LLM performance on certain benchmarks and previous studies on GPT models, are outdated. We will also update these sections with more recent literature.
>
> We agree that the interpretation of workload distribution can be refined and will revise the text to more accurately describe the variability of participant perceptions. We will change the text to the following to reflect the reviewer’s comment:
>
> Our empirical results are consistent with intuition: software developers need to possess AI competencies within software development, which indicates the necessity of equipping AI-specific skills, such as prompt engineering, critical evaluation of AI outputs, and AI-assisted coding workflow management in their development practice.
>
> [11] We also appreciate the suggestion to elaborate the discussion on “effective human-AI collaboration”, we could not develop more discussion on this mainly due to the space limitation. In the revised version, we will explicitly list it as a future research direction, encouraging researchers to conduct deeper examinations.
>
> Regarding the other comments section of the review, in our camera-ready version,
> - We will rephrase ChatGPT to GPT models.
>
> - We will rephrase the section title to Measuring AI’s Code Quality.
>
> - As the goal of the paper was to sample software managers generally, we did not include or exclude respondents based on their company. Given this, we will rephrase “sampling software managers employed across a range of companies” to “sampling software managers employed across both tech-focused and non-tech-focused companies.
>
> - The original paper utilized AI to refer to decision-making AI tools. We will include this detail within our camera-ready paper.
>
> - We will clearly define the term “Knowledge Work” within the Key Terms section as “work dominated by cognitive effort, in which workers use specialized knowledge, judgment, and problem-solving to create, apply, or share knowledge-based outputs [1].”  (Ref [1]: El-Farr, Hadi K. "Knowledge work and workers: a critical literature review." Leed University Business School, Working Paper Series 1, no. 1 (2009): 1-15. )
>
> We thank the reviewer again for the constructive suggestions, which we believe will significantly improve the clarity and impact of the paper.